# Influenza and Other Prophylactic Vaccination Coverage in Polish Adult Patients Undergoing Allergen Immunotherapy—A Survey Study among Patients and Physicians

**DOI:** 10.3390/vaccines10040576

**Published:** 2022-04-08

**Authors:** Ewa Czerwińska, Marita Nittner-Marszalska, Janusz Zaryczański, Grzegorz Gąszczyk, Agnieszka Mastalerz-Migas, Leszek Szenborn

**Affiliations:** 1Clinical Department of Paediatrics and Infectious Diseases, Wroclaw Medical University, 50-368 Wroclaw, Poland; leszek.szenborn@umed.wroc.pl; 2Clinical Department of Internal Medicine, Pneumology and Allergology, Wroclaw Medical University, 50-369 Wroclaw, Poland; marmarsz@gmail.com; 3Department of Pediatrics, University Clinical Hospital in Opole, 45-040 Opole, Poland; januszz@wcm.opole.pl; 4Department of Allergology, Medical Centre in Karpacz, 58-540 Karpacz, Poland; gaszczyk1602@gmail.com; 5Department of Family Medicine, Wroclaw Medical University, 51-141 Wroclaw, Poland; agnieszka.mastalerz-migas@umed.wroc.pl

**Keywords:** influenza vaccine, allergen immunotherapy, prophylactic vaccines, vaccine coverage

## Abstract

Vaccines against infectious diseases may raise safety concerns in patients undergoing allergen immunotherapy (AIT). The objective of our study was to investigate influenza vaccine and other selected prophylactic vaccines coverage in patients treated with AIT and the attitude of physicians towards vaccinations in this group of patients. We conducted a questionnaire-based study among patients undergoing AIT and physicians. The patients’ survey evaluated influenza and other prophylactic vaccines coverage. The physicians’ survey assessed their experience and opinions on prophylactic vaccinations during AIT. In total, 176 patients (aged 18–79 years) and 120 doctors filled the questionnaires. Patients were assigned to two groups—inhaled allergens group (*n* = 101) and insect venoms group (*n* = 68). The number of patients who received any dose (36% and 45%, *p* = 0.26), as well as two or more doses (17% and 22%, *p* = 0.43) of influenza vaccine was comparable between two groups. However, in both groups there was a significant (*p* < 0.0001) decrease in influenza vaccine uptake after the beginning of AIT. Patients from the inhaled allergens group declared a higher tetanus vaccine rate (41% vs. 19%, *p* = 0.004). The groups did not differ in the pneumococcal and tick-borne encephalitis vaccination coverage. A majority of doctors believe that prophylactic vaccinations in patients undergoing AIT are safe and effective (96% and 94%, respectively); however, as many as 87% of them identify with the need to create clear recommendations regarding vaccinating patients undergoing AIT. Prophylactic vaccine coverage is not satisfactory among Polish adult patients undergoing AIT. Polish doctors are convinced of the validity of prophylactic vaccinations during AIT.

## 1. Introduction

The invention and worldwide dissemination of vaccinations against infectious diseases is one of the greatest successes in medicine. Vaccines help to prevent contagious diseases and save the lives of many people every year [1]. Most of the prophylactic vaccines require completing only a primary schedule, typically in childhood, but some of them need to be repeated in later life (e.g., an annual influenza vaccine). These medical procedures are proven to be safe and are also recommended for use for patients with different concomitant diseases [2].

A special population consists of allergic patients undergoing allergen immunotherapy (AIT). Currently, it is the only casual treatment available for allergic patients [3]. Inducing immunological tolerance requires regular administration of allergen vaccines—every day in the case of sublingual immunotherapy (SLIT) [4], or on a monthly basis in the case of subcutaneous immunotherapy (SCIT) [5]. Both types of allergen immunotherapy, and particularly in sublingual form, are safe procedures, and adverse reactions are mainly local and tend to be mild [6]. Prophylactic vaccinations in patients with different allergies are also considered to be secure [7].

The guidelines of the European Academy of Allergy and Clinical Immunology (EAACI) recommend separating the administration of allergen extracts and vaccinations against infectious diseases by at least one week [8]. A summary of product characteristics (SmPC) of some allergen vaccines intended for subcutaneous use advise maintaining different time intervals between the two types of vaccines as well. However, the experience of doctors and the literature indicate the safety of administering prophylactic and therapeutic (SCIT) vaccines in a short time interval (even on the same day) [9,10]. For everyday medical practitioners, these findings may increase concerns about the appropriate approach to AIT patients and prophylactic vaccines.

Vaccine acceptance and patients’ own motivation to get vaccinated play important roles in reaching satisfactory vaccine uptake in the population. Unfortunately, a strong tendency to discredit the validity and safety of prophylactic vaccinations has been observed lately. As a consequence, immunization rates of different vaccines have decreased. This results mainly from disinformation spread by anti-vaccination movements [11]. Owing to the different causes of vaccine hesitancy, various and tailored methods of improving vaccination acceptance are sought and need to be implemented [12]. This problem appears particularly important within the context of COVID-19—achieving herd immunity due to vaccinations is necessary to gain control over the pandemic [13,14].

Since prophylactic vaccines remain a widely discussed subject in a public debate, many studies regarding vaccinations in various groups of patients (depending on their age, concomitant diseases, residence or occupation) have been conducted. However, the problem of managing prophylactic vaccines in patients undergoing allergen immunotherapy has not been thoroughly studied. Moreover, little is known about the attitude of physicians regarding this topic.

The aim of our study was to asses vaccine coverage in patients undergoing allergen immunotherapy and the experience of doctors of combining vaccinations against infectious diseases and allergen immunotherapy.

## 2. Materials and Methods

### 2.1. Participants

#### 2.1.1. Patients

Adult patients undergoing AIT who are treated in two allergology ambulatory care clinics in different Allergology Departments in Poland (Opole and Karpacz) and who agreed to take part in the study.

#### 2.1.2. Physicians

Physicians who take care of allergic patients (paediatricians, allergists, general practitioners).

### 2.2. Questionnaires

For our study, we developed two original Polish-language surveys. They were completely anonymous (thus we did not obtain written informed consent from study participants). Both questionnaires were evaluated by six physicians to ensure the accuracy of the questions. 

In the result presentation, the exact number of responses was taken into consideration, which was not always equal to the number of a particular group population.

English translation of the surveys is presented as an attachment to this paper (Appendix A).

#### 2.2.1. Patients’ Questionnaire

The questionnaire was offered during the first half of the year 2021 to adult patients undergoing AIT.

The patients were asked to fill in printed versions of the questionnaire before a scheduled visit with a doctor. The survey evaluated the type of allergy, adverse reactions after allergen immunotherapy, influenza and other vaccines coverage and adverse events after receiving an influenza vaccine.

#### 2.2.2. Physicians’ Questionnaire

The second questionnaire was conducted in a group of physicians during the entirety of 2021. The questionnaires were sent to the potential respondents via telecommunication methods (e-mails, social media) and offered in a printed version during conferences. The survey assessed the rules followed to schedule vaccinations in AIT patients, as well as opinion on the safety and effectiveness of combining vaccinations against infectious diseases and allergen immunotherapy.

### 2.3. Statistical Analysis

Statistical significance among groups was calculated by the Fisher’s exact test. Values of *p* < 0.05 were considered significant.

### 2.4. Ethical Approval

The study was approved by the Ethics Committee of Wroclaw Medical University.

## 3. Results

### 3.1. Patient’s Questionnaire

#### 3.1.1. Baseline Characteristics of the Group

We received 176 completed questionnaires. The response rate was 100% among patients from Opole and 54% among patients from Karpacz. To evaluate differences between the patients, we divided them into groups depending on the type of received immunotherapy—inhaled allergens (101 patients), insect venom (68 patients) or both (seven patients). As the group of patients undergoing both inhaled allergens and insect venom immunotherapy consisted only of seven patients, we excluded it from further analysis.

The mean age of patients was 38 years old. All patients were undergoing SCIT (*n*= 168) and one patient was undergoing both SCIT and SLIT. We asked about having children as cohabitants as it can be a risk of repeated exposure and the increased incidence of infectious diseases (e.g., seasonal influenza); this problem affected almost 70% of the patients from insect venoms group.

The demographic and clinical data of both groups is presented in Table 1. 

#### 3.1.2. Influenza Vaccine Coverage before and during AIT along with Influenza Vaccine Adverse Events

When compared between the two groups, patients did not differ in influenza vaccine uptake in the past along with after the beginning of allergen immunotherapy. However, in both groups there was a significant (*p* < 0.0001) decrease in influenza vaccine uptake after the beginning of AIT.

There was no difference between influenza vaccine coverage among patients from both groups suffering from concomitant diseases in comparison to those who did not declare concomitant diseases predisposing them to a severe course of influenza (asthma, diabetes), *p* = 0.445.

Allergic reactions after influenza vaccine administration (defined as allergic rhinitis/conjunctivitis, dyspnoea, urticaria, anaphylactic shock), were reported only by patients from the insect venom group. Local adverse reactions to the influenza vaccine were more frequent in the inhaled allergens group (49% vs. 23%, *p* = 0.043), while there was no difference in the occurrence of non-allergic systemic reactions between the two groups of patients.

The data is presented in Table 2.

#### 3.1.3. The History of Other Vaccinations in Adulthood (Irrespective of AIT) and Attitude towards Vaccines against COVID-19 among Patients Undergoing Allergen Immunotherapy

The most popular vaccines among Polish adults undergoing AIT were the tetanus and pertussis vaccines. Declared tetanus vaccination coverage was higher among patients from the inhaled allergens group (44% vs. 21%, *p* = 0.006). Surprisingly, 70% of patients from insect venoms group declared being vaccinated against pertussis, with only around 20% being vaccinate against tetanus. These results are probably overestimated regarding the pertussis vaccine or underestimated regarding tetanus vaccine as the only vaccinations against pertussis registered in Poland are combined with diphtheria and tetanus toxoids, while vaccination against tetanus in adult patients can be performed with the use of monovalent tetanus vaccines or combined vaccines with diphtheria, pertussis and polio components (bivalent, trivalent or quadrivalent formulations). The groups did not differ significantly in the pneumococcal vaccination and tick-borne encephalitis vaccination coverage; however, the declared uptake of these two vaccines was low. In both groups the will to get vaccinated against COVID-19 was expressed by around 50% of respondents.

The data is presented in Table 3.

### 3.2. Doctors’ Questionnaire

#### 3.2.1. Characteristics of the Group

We received 120 filled questionnaires from medical professionals. Respondents represented all age groups. Most doctors were specialists in paediatrics, allergology or primary care. 18% of doctors declared being specialists in at least two different medical specialties. The basic characteristics of the group are presented in Table 4.

#### 3.2.2. General Information about the Patients Being under Care of Surveyed Doctors

The majority of doctors (91%) admitted being in charge of patients who are undergoing allergen immunotherapy. Most doctors (63%) in their practice deal with patients undergoing both types of allergen immunotherapy (SCIT and SLIT), while the minority of doctors (8%) only take care of patients undergoing sublingual immunotherapy. As many as 93% of doctors perform vaccinations against infectious diseases in their everyday practice (detailed data in Table 5).

#### 3.2.3. Experience in Conducting Vaccinations against Infectious Diseases in Patients Undergoing Allergen Immunotherapy

The majority of doctors (74%) responded that the knowledge about treatment with allergen immunotherapy in a particular patient leads to more conscious vaccination schedule planning. The most common answers regarding the rules obeyed while planning a vaccination schedule in AIT patients were: keeping a one week interval between therapeutic and prophylactic vaccinations, following allergologists recommendations and adhering to the summary of product characteristics of a particular allergen extract. Respondents had the possibility to choose more than one answer while responding to this question and numerous different answers may show a lack of certainty about the best response. The vast majority of doctors believe that vaccinating patients undergoing AIT is safe and effective (96% and 94%); most of them recommend that their patients get annual vaccinations against influenza. As many as 87% of respondents agree that there is a need for creating clear recommendations on vaccinating patients undergoing AIT (data in Table 6).

## 4. Discussion

There were three major findings arising from our survey-based study. Firstly, we demonstrated that influenza vaccine coverage is suboptimal, does not differ depending on the type of allergen immunotherapy the patient is undergoing (inhaled allergens vs. insect venoms), and decreases after the beginning of allergen immunotherapy. Secondly, general vaccine coverage among adult patients undergoing AIT is not satisfactory. Finally, Polish doctors taking care of AIT patients are convinced about the safety and effectiveness of prophylactic vaccines in allergic patients undergoing AIT, but planning vaccination visits is problematic, as they lack clear recommendations addressing the time interval between the procedures.

### 4.1. Influenza Vaccine Coverage among Patients Undergoing AIT

The influenza vaccine is the best method of preventing this viral disease [15], therefore it is recommended for all Polish citizens over six months of age due to epidemiological reasons, and it is particularly advised for patients over 55 years old or with chronic disorders, like asthma or diabetes [2].

We have demonstrated that around 40% of patients undergoing AIT claim to have received at least a single influenza vaccine during adulthood before the beginning of AIT, which corresponds with findings from a Polish nationwide survey (32%) conducted in 2013 [16]. These numbers are not satisfactory, as the influenza vaccine composition changes every year, thus a single vaccination is not sufficient—the vaccine is recommended for annual use. Among our patients, only around 20% declared being vaccinated against influenza more than once. Influenza vaccine hesitancy in the general population is connected with a belief about inadequate influenza vaccine effectiveness, the perceived low possibility of contracting influenza, and concerns about adverse events [16,17]. This topic has yet to be investigated among AIT patients.

According to data published by the ECDC (European Centre for Disease Prevention and Control), influenza vaccine uptake in the entire Polish population is lower than 4% (seasons 2015/2016 and 2016/2017) [18]. Similar data were collected from Polish nationwide survey studies carried out in 2013 and 2016 which found that the declared influenza vaccination rate was approximately 6–7%, with another 6–7% of patients willing to get vaccinated later in the influenza season [16,17]. Observations made in 2015 on Polish patients suffering from chronic diseases suggest that influenza vaccination coverage may be higher (9–58%) depending on the patient’s medical condition (with the highest rate reported by patients with chronic pulmonary diseases) [19].

The other problem is the noticeable decrease in the interest in influenza vaccines after the beginning of AIT (36% and 45% previous to, and 14% and 6% after the beginning of AIT, respectively). These results suggest that the regular administration of an allergen extract may impede the performing of prophylactic vaccinations against influenza. Furthermore, our results reveal that even less patients decided to get vaccinated against influenza during the ongoing COVID-19 pandemic (3–9%). These findings correlate with the survey conducted in November 2020 on the general population of Polish adults, which found that at that time only 5.5% of patients had already been vaccinated against influenza [20]. It is alarming as to why influenza vaccine coverage is so low among Poles, while according to the recent (2020) State of Vaccine Confidence in the European Union and United Kingdom, as many as 78.1% and 82.4% of Polish respondents agree that the seasonal influenza vaccine is important and safe, respectively [11].

Over 20% of our patients belong to the group at high risk of severe influenza and its complications because of their age or concomitant diseases, which makes them candidates for the annual influenza vaccine uptake. As it results from the outcomes of our survey, the past history of vaccinations against influenza in this special group of patients is not significantly higher than in general population.

The presented data regarding influenza vaccine coverage are still not satisfactory and much lower than recommended by different health organizations, for example the target of an annual 70% coverage for the population over six months of age (USA Healthy People 2030) [21], or of 75% coverage of elderly patients (over 65 years old) in the European Region (WHO, ECDC) [22,23].

### 4.2. General Vaccine Coverage among Patients Undergoing AIT

According to the WHO, one of the ten threats to global health is vaccine hesitancy [24]. It might seem that the problem concerns only some of the vaccinations that are mandatory for use in children (e.g., MMR vaccine). In fact, although the numbers may vary between countries and vaccine type, the general vaccine coverage among adult patients is low [25,26,27].

In the group of our respondents, the highest vaccination coverage applies to tetanus and pertussis vaccines. This may result from the fact that in Poland these vaccines are recommended for all adult patients—a combined dTap vaccine every ten years starting at age 19, during every pregnancy and in all people who have or may have contact with infants. Additionally, the tetanus vaccine (with or without the pertussis component) should be considered in all patients after exposure to tetanus, depending on the history of tetanus vaccinations [2]. Surprisingly, in one of our groups, 70% of patients declared being vaccinated against pertussis, although only 21% were vaccinated against tetanus. These results are probably overestimated regarding the pertussis vaccine or underestimated regarding the tetanus vaccine as the only vaccinations against pertussis registered in Poland are combined with diphtheria and tetanus toxoids. These unlikely results can indicate another problem, which is patients’ low awareness of their own vaccination history. One of the solutions to this issue could be offering adult patients vaccination booklets similar to those prepared for paediatric patients or the implementation of electronic vaccination tracking.

Pneumococcal infections in adults are connected with high mortality, particularly among the elderly, patients with chronic respiratory diseases (including asthma), diabetes mellitus, chronic heart disease and smokers [28,29]. According to the Polish vaccination schedule, the pneumococcal vaccine is recommended for all adult patients with various chronic disorders, impaired immune function and smokers [2]. In our patients, the vaccination rate is low (5–7%), which may result from the relatively low mean of the age of the respondent population (38 years old).

Poland belongs to a tick-borne encephalitis (TBE) risk region (the “TBE belt”); however, the degree of endemicity varies depending on the part of the country [30]. The Polish vaccination schedule recommends vaccination against TBE to all people in these endemic regions and who work or spend time outdoors (and therefore are at risk of a tick bite) [2]. In our patients, the vaccination rate is lower than 5%. Survey studies from other “TBE belt” countries indicate higher vaccination coverage [31].

Vaccines against COVID-19 are the hope for ending the COVID-19 pandemic. As these vaccines rely on new technologies (mRNA and viral vectors) and were introduced for public use relatively quickly, many patients have concerns regarding their safety and efficacy [32,33]. Among our respondents, around 50% were willing to get vaccinated against COVID-19 (the study was conducted before and at the beginning of COVID-19 mass vaccinations in Poland). We did not have the chance to verify if those respondents actually got vaccinated against COVID-19, but after a year of accessibility of different vaccines against COVID-19 in Poland, the number of fully vaccinated Polish citizens is approximately 59% (figures as of 22 February 2022) [34].

Vaccine acceptance, including the COVID-19 vaccine, is crucial in guaranteeing herd immunity and in eliminating or decreasing the prevalence of infectious diseases [35,36]. Vaccine acceptance depends on many factors connected with the specific vaccine, the perception of a particular disease among the population and the general trust in science and the healthcare system [35]. During the ongoing pandemic, many studies regarding attitudes towards COVID-19 vaccines have emerged. Similarly to other vaccinations, beliefs and concerns related to COVID-19 vaccines are connected with age, gender, residency, perceived risk of the disease, education level, income, life circumstances (pregnancy) or access to social media [37,38,39,40,41,42,43]. For this reason, we want to underline the importance of adjusting the means of communication to individual needs, as such interventions are more effective than the ones addressed to the whole population. Since patients undergoing AIT are not homogenous in terms of demographic or economic factors, the message concerning the possibility and safety of vaccinations, including the COVID-19 vaccine, should be tailored to a particular patient.

It is worth noting that 55% of Polish respondents declare overall vaccine confidence (comparing to 37% in 2018) [11]. On the contrary, among patients who responded to our questionnaire, different vaccines’ coverage is rather low. Unfortunately, there are no official data available for adult vaccination coverage in Poland, as all information regarding this topic derive from individual surveys, which are usually conducted on patients suffering from various chronic diseases or special groups of patients and not on the general public [19,44,45]. Thus, it is hard to determine if low vaccine coverage among our respondents results only from the fact of undergoing AIT or the problem is a more complex one, especially if only individual patients declared having problems with getting vaccinations while undergoing AIT.

### 4.3. Doctors Attitude

Studies regarding vaccine hesitancy indicate the important role of doctors, especially general practitioners, on shaping patients’ attitudes about immunization. Fortunately, many patients still indicate that their doctors are the most trusted source of health information [25,44,46]. This should prompt the doctors to update their knowledge and to provide the patients with information compliant with evidence-based medicine. On the other hand, some research points to limited knowledge about adult vaccinations among healthcare providers [47,48,49].

The majority of doctors who took part in our questionnaire agreed on the safety and effectiveness of vaccinations among patients undergoing AIT (96% and 94%, respectively). Furthermore, over 80% of respondents recommend that their patience are vaccinated against influenza annually. Unfortunately, this does not lead to satisfactory vaccine coverage.

Our results correlate with the opinion on the lack of negative interference between vaccinations and AIT declared by 95% of AIT experts who took part in an international survey. The majority of doctors did not observe any alarming AIT (98%) or prophylactic vaccine (87%) adverse effects when combining these two procedures, and the only reported unfavourable reactions were local and mild [50].

Various answers regarding the rules obeyed while planning the vaccination schedule in AIT patients and the opinion of more than 90% of doctors that clear recommendations concerning vaccinations among AIT patients are needed may indicate that doctors taking care of such patients are faced with a technical problem of arranging a proper vaccination date. More attention should be paid to simplifying the process of vaccinating patients undergoing AIT, and such recommendations should be prepared in the local language by local societies of allergology and vaccinology.

The problem of combining allergen immunotherapy and prophylactic vaccinations in adult patients has probably not been analysed yet, as we did not find any publications regarding this topic. Since there are no studies assessing both vaccine coverage in adult patients undergoing AIT and the attitude of physicians towards vaccinating such patients, we would like to emphasize the novelty and significance of our research. We hope that it will contribute to increasing the uptake of prophylactic vaccines among adult patients undergoing AIT, particularly those who are at risk of infectious diseases due to their age, concomitant diseases or place of residence. The group that would benefit the most from simplifying the vaccination process are certainly AIT patients themselves, however it would also improve protection in the general population due to herd immunity.

## 5. Study limitations

We are aware of certain limitations of our study, as questionnaire-based studies are always connected with subjective responses. Firstly, some of the reported symptoms (allergic reactions during AIT, adverse reactions after vaccines against infectious diseases) may be tendentious as they were not measured by the same person using identical methods. Secondly, the percentage of different vaccines administered to the patients was self-reported, based on patients’ memory and not on medical records (which can explain the differences in the percentage of patients from the insect venom group reporting being vaccinated against tetanus and pertussis). We believe, however, that the study reflects the current state of influenza and other vaccines’ uptake in Polish adult patients undergoing AIT.

## 6. Conclusions

Our findings indicate that prophylactic vaccine coverage is not satisfactory among Polish adult patients undergoing allergen immunotherapy. Further educational measures need to be implemented to increase knowledge and to promote the benefits of vaccinations against infectious diseases among these patients.

Polish doctors are convinced about the safety and effectiveness of prophylactic vaccinations during allergen immunotherapy, but they lack clear recommendations regarding scheduling vaccination visits in patients undergoing AIT. There is a need for better communication between both surveyed groups to increase patients’ vaccine confidence and vaccine coverage, particularly in high risk groups.

## Figures and Tables

**Table 1 vaccines-10-00576-t001:** Baseline characteristics of the patients.

	Inhaled Allergens Group (*n* = 101)	Insect Venoms Group (*n* = 68)
Age (years)		
18–29	30 (29.7%)	17 (25.4%)
30–39	36 (35.6%)	14 (20.9%)
40–49	24 (23.8%)	13 (19.4%)
50–59	9 (8.9%)	15 (22.4%)
60–69	1 (1%)	8 (11.9%)
70–79	1 (1%)	0
>80	0	0
		(1—unspecified)
Type of AIT		
only SLIT	0	0
only SCIT	100	68
SLIT and SCIT	1	0
Allergic adverse reactions to AIT:		
- urticaria	1%	1.50%
- allergic rhinitis/conjunctivitis	5%	1.50%
- dyspnoea	1%	15%
- anaphylactic shock	0	1.50%
Concomitant diseases:		
- in general, including:	29%	38.00%
- asthma	75.90%	3.90%
- hypertension	13.80%	53.80%
- diabetes	0%	34.60%
- asthma, hypertension	10.30%	3.90%
- asthma, diabetes	0%	3.90%
Children < 14 years old as cohabitants	24%	69%

**Table 2 vaccines-10-00576-t002:** Influenza vaccine coverage before and during AIT along with influenza vaccine adverse events.

	Inhaled Allergens Group (*n* = 101)	Insect Venoms Group (*n* = 68)	*p*
Vaccinations against influenza in the past (general)	36%	45%	0.26
Vaccinations against influenza in the past—twice or more	17%	22%	0.43
Any vaccination against influenza in the past among patients with asthma/diabetes	58%	40%	0.482
Vaccinations against influenza during AIT	14%	6%	0.131
Vaccination against influenza during 2019/2020 season	10%	6%	0.569
Vaccination against influenza during 2020/2021 season	9%	3%	0.206
Systemic allergic reactions after influenza vaccine administration	0%	10%	0.093
Local reactions after influenza vaccine administration	49%	23%	0.043
Systemic reactions (other than allergic) after influenza vaccine administration	26%	27%	1.000

**Table 3 vaccines-10-00576-t003:** The history of other vaccinations in adulthood (irrespective of AIT) and attitude towards vaccines against COVID-19 among patients undergoing allergen immunotherapy.

	Inhaled Allergens Group (*n* = 101)	Insect Venoms Group (*n* = 68)	*p*
Tetanus vaccination	44%	21%	0.006
Pneumococcal vaccination	7%	5%	0.741
Pertussis vaccination	36%	70%	<0.0001
Tick-borne encephalitis vaccination	4%	3%	1.000
A will to vaccinate against COVID-19	54%	52%	0.872
Any problems with having vaccinations against infectious diseases because of AIT	3%	1,50%	1.000

**Table 4 vaccines-10-00576-t004:** Age and professional education of responding doctors.

	Medical Professionals Respondents (*n* = 120)
Age:	
<29	9 (8%)
30–39	40 (33%)
40–49	35 (29%)
>50	36 (30%)
Education: (multiple choice question)	
primary care resident	15 (13%)
primary care specialist	15 (13%)
paediatrics resident	15 (13%)
paediatrics specialist	51 (43%)
internal medicine resident	1 (1%)
internal medicine specialist	13 (11%)
allergology specialist	29 (24%)
other	8 (7%)
two or more medical specialties	21 (18%)

**Table 5 vaccines-10-00576-t005:** General information regarding patients undergoing therapeutic (AIT) and prophylactic vaccinations that are under the care of the surveyed doctors.

	Medical Professionals Respondents (*n* = 120)
Frequency of taking care of patients undergoing AIT:	
more often than once a week	46 (38%)
once a week—once a month	21 (18%)
less often than once a month	44 (37%)
never	9 (8%)
AIT route of administration in patients being under care of surveyed doctors:	
only SCIT	32 (29%)
only SLIT	9 (8%)
SCIT and SLIT	70 (63%)
Age group of patients vaccinated against infectious diseases in everyday practice of surveyed doctors:	
children	41 (34%)
adults	21 (18%)
both	51 (43%)
none	7 (6%)

**Table 6 vaccines-10-00576-t006:** Attitude towards safety and effectiveness of prophylactic vaccines in patients undergoing AIT; experience of combining AIT and vaccinations against infectious diseases, particularly influenza vaccine.

	Medical Professionals Respondents (*n* = 120)
The impact of AIT on planning vaccinations against infectious diseases:	
vaccination planning is more aware	89 (74%)
vaccination planning is more difficult	10 (8%)
there is no impact	21 (18%)
An interval applied between AIT and vaccinations against infectious diseases: (multiple choice question)	
one week (AIT—one week—prophylactic vaccine OR prophylactic vaccine—one week—AIT)	68 (36%)
according to the SmPC * of the allergen extract	36 (19%)
according to the SmPC * of the vaccine	31 (16%)
according to allergologist suggestions	45 (24%)
regardless of the interval	6 (3%)
no vaccinations during AIT	3 (2%)
Opinion on vaccines safety during AIT: (multiple choice question)	
vaccines are safe, there are studies confirming this thesis	36 (29%)
vaccines are safe, there are recommendations allowing vaccinating patients during AIT	85 (67%)
vaccines are not safe during AIT	5 (4%)
Opinion on vaccine effectiveness during AIT:	
vaccines are effective	113 (94%)
vaccines have limited effectiveness	0
vaccines are not effective	0
never considered this topic	7 (6%)
Recommending to get vaccinated against influenza every year:	
only patients undergoing subcutaneous AIT	0
only patients undergoing sublingual AIT	0
all patients undergoing AIT, regardless of the route of administration	102 (85%)
only patients undergoing AIT with risk of poor outcome of influenza	15 (12.5%)
I don’t recommend vaccines against influenza to AIT patients	3 (2.5%)
Opinion on the need for clear recommendations on vaccinating patients undergoing AIT:	
yes, there is a need	104 (87%)
no, current recommendations are sufficient	16 (13%)

* summary of product characteristics.

## Data Availability

Study database is available on request at the Department of Pediatric Infectious Diseases, Wroclaw Medical University.

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
