# Peer review of "Influenza and Other Prophylactic Vaccination Coverage in Polish Adult Patients Undergoing Allergen Immunotherapy—A Survey Study among Patients and Physicians"

_vaccines, 2022, doi:10.3390/vaccines10040576_

Round 1
Reviewer 1 Report
This is purely a survey study but may provide important information about the vaccination pattern and nature among public especially against influenza.
I urge why the author have not considered COVID-19 vaccination patter. If not at least dedicate few lines citing the importance of COVID-19 cases and it vaccination pattern based on which the current study is relevant and done.
Ethical concern: How the article will be published if written consent was not obtained?
Materials method section needs to be carefully divided into distinct parts.
Why you think that the opinion of 6 physicians will be sufficient for this study in relation to numerous patients?
Why opinions of immunotherapists only was not considered.
Reference section is little poor, along with above, some more recent references need to be cited.
Author Response
REVIEWER 1 COMMENTS:
- COMMENT 1: This is purely a survey study but may provide important information about the vaccination pattern and nature among public especially against influenza.
RESPONSE: Thank you for your comment.
- COMMENT 2: I urge why the author have not considered COVID-19 vaccination patter. If not at least dedicate few lines citing the importance of COVID-19 cases and it vaccination pattern based on which the current study is relevant and done.
RESPONSE: We did not consider COVID-19 vaccination pattern, as when our study was conducted, COVID-19 vaccines were not available for whole Polish population yet. However, we agree that COVID-19 vaccines are of great importance for public health, thus we have added a new paragraph concerning this topic in lines 335-348.
- COMMENT 3: Ethical concern: How the article will be published if written consent was not obtained?
RESPONSE: Formal written informed consent was not required because we used data from anonymous questionnaire. The patients and health-care professionals who participated in the survey cannot be recognized basing on their responses. Participation in this study was voluntary, the submission of responses implied an agreement to participate.
- COMMENT 4: Materials method section needs to be carefully divided into distinct parts.
RESPONSE: Thank you for this comment. We have amended Matherials and Methods section according to yours suggestions to make it more accurate.
- COMMENT 5: Why you think that the opinion of 6 physicians will be sufficient for this study in relation to numerous patients?
RESPONSE: The physicians who evaluated our questionnaires are experts in the field of allergology (including allergen immunotherapy), vaccinology or family medicine, have considerable experience in working with patients (hence know how to clarify and simplify their language to suit patients’ possibility of understanding) and were authors of different survey studies. We believe that their abilities and opinions have helped to create adequate questionnaires.
- COMMENT 6: Why opinions of immunotherapists only was not considered.
RESPONSE: We included opinions of physicians of different specialties, as in Poland the main role in vaccines administration attributes to family medicine doctors, paediatricians and internal medicine specialists. As allergologists, particularly allergen immunotherapy specialists, rarely perform prophylactic vaccinations in patients, their role relies mainly on encouraging and reminding patients about recommended vaccinations. For that reason, they are an important chain in building vaccine acceptancy in patients with different allergic disorders. We included results of a current survey among allergen immunotherapy experts regarding management of AIT and vaccinations which are consistent with our observations (lines 372-376, reference 50).
- COMMENT 7: Reference section is little poor, along with above, some more recent references need to be cited.
RESPONSE: There are not many articles regarding the topic of vaccinating patients undergoing allergen immunotherapy, thus we are not able to provide more recent research than the ones mentioned in our discussion. However, according to suggestions, we have added a paragraph about the importance of COVID-19 vaccinations (lines 335-348, references 35-43) and an article representing the opinion of allergen immunotherapy specialists regarding prophylactic vaccinations (lines 372-376, reference 50).
Reviewer 2 Report
This work aims to evaluate any variation in vaccination coverage (prophylactic vaccines such as influenza, tetanus, etc.) in adult patients undergoing anti-allergic de-sensitizing therapy. The point of view of patients and administrators is analyzed through the administration of (different) questionnaires for the two groups.
The 1st questionnaire with the aim of evaluating any changes in the coverage rate of prophylactic vaccines (influenza and others) in the course of desensitizing therapy.
The 2nd questionnaire (for ADMINISTRATORS / PRESCRIPTORS) to evaluate their opinions and collect their experiences in this regard.
Patients 176 divided into 2 groups:
1) 101 patients undergoing desensitizing vaccine against inhalant allergens (pollen, molds, epidermal derivatives)
2) 68 patients subjected to desensitizing vaccine against hymenoptera venom
NB: the 7 missing patients had both vaccines in progress
Doctors 120
RESULTS
- Influenza vaccination coverage was sub-optimal overall and further lower than the general population coverage for patients undergoing desensitization. but there is no significant difference between the two patient groups (those with desensitization for inhalants and for hymenoptera venom).
- The Polish doctors who prescribe the desensitizing immune therapy are however convinced of the safety and efficacy of vaccine prophylaxis to be carried out on subjects who are undergoing desensitizing therapy, however they underline the need for the creation of Guidelines and clear specific recommendations.
The numbers are limited, but it is not easy to find an adequate number of subjects who are undergoing desensitization and who need a prophylactic vaccination in the same period.
The use of the questionnaire means has obvious limits connected to the subjectivity and "memory" of the respondent (even the authors admit this).
Author Response
REVIEWER 2 COMMENTS:
- COMMENT 1: This work aims to evaluate any variation in vaccination coverage (prophylactic vaccines such as influenza, tetanus, etc.) in adult patients undergoing anti-allergic de-sensitizing therapy. The point of view of patients and administrators is analyzed through the administration of (different) questionnaires for the two groups.
The 1st questionnaire with the aim of evaluating any changes in the coverage rate of prophylactic vaccines (influenza and others) in the course of desensitizing therapy.
The 2nd questionnaire (for ADMINISTRATORS / PRESCRIPTORS) to evaluate their opinions and collect their experiences in this regard.
Patients 176 divided into 2 groups:
1) 101 patients undergoing desensitizing vaccine against inhalant allergens (pollen, molds, epidermal derivatives)
2) 68 patients subjected to desensitizing vaccine against hymenoptera venom
NB: the 7 missing patients had both vaccines in progress
Doctors 120
RESULTS
- Influenza vaccination coverage was sub-optimal overall and further lower than the general population coverage for patients undergoing desensitization. but there is no significant difference between the two patient groups (those with desensitization for inhalants and for hymenoptera venom).
- The Polish doctors who prescribe the desensitizing immune therapy are however convinced of the safety and efficacy of vaccine prophylaxis to be carried out on subjects who are undergoing desensitizing therapy, however they underline the need for the creation of Guidelines and clear specific recommendations.
RESPONSE: Thank you for your comment. It briefly summarizes our study and its results.
- COMMENT 2: The numbers are limited, but it is not easy to find an adequate number of subjects who are undergoing desensitization and who need a prophylactic vaccination in the same period.
RESPONSE: That’s correct. The number of patients that require, simultaneously, a prophylactic vaccination and desensitization is relatively small. However, the problem arises seasonally when anti-flu vaccinations overlap with immunotherapy, newly initiated or continued, concerning mostly adults. Additionally, in the last two years, we have had to consider administering anti-COVID-19 vaccinations with current immunotherapy. We are afraid that the issue may recur in the years to come, which is why we are of the opinion that the problem is worth addressing.
- COMMENT 3: The use of the questionnaire means has obvious limits connected to the subjectivity and "memory" of the respondent (even the authors admit this).
RESPONSE: We agree and are aware of the limitations of our research which result from the character of survey studies.
Reviewer 3 Report
Dear authors,
thank you for this study. It is an important topic, and the paper is well written. I dont have many remarks:
Authors try to map the perceptions of physicians regarding vaccination inPoland.
the study is relevant as physicians have to convince the patients of
the vaccination. the paper add the limited evidence. the conclusions are consistent with the evidence.the references are appropriate
1. Remarkable is 100% response rate in 1 hospital and 50% in the other. How can this be explained?
2. Sampling of physicians could be improved. This is why I asked if
pediatricians are the correct sample for this research. While the average is 38, the majority of doctors interviewed are paeditricians. How representative is this population compared to other disciplines?
Author Response
REVIEWER 3 COMMENTS:
- COMMENT 1: Dear authors,
thank you for this study. It is an important topic, and the paper is well written. I dont have many remarks:
Authors try to map the perceptions of physicians regarding vaccination in Poland.
the study is relevant as physicians have to convince the patients of
the vaccination. the paper add the limited evidence. the conclusions are consistent with the evidence.the references are appropriate
RESPONSE: Thank you for this comment.
- COMMENT 2: 1. Remarkable is 100% response rate in 1 hospital and 50% in the other. How can this be explained?
RESPONSE: Thank you for pointing this out. We agree that there are discrepancies in response rate between two towns. In each ambulatory care clinic there was a different doctor responsible for collecting questionnaires from the patients; various response rates result from the possibility to ensure returning the questionnaires by patients and patients’ will to take part in the study.
- COMMENT 3: 2. Sampling of physicians could be improved. This is why I asked if
pediatricians are the correct sample for this research. While the average is 38, the majority of doctors interviewed are paeditricians. How representative is this population compared to other disciplines?
RESPONSE: We agree that sampling of physicians could be improved. However, our aim was to collect the responses from doctors of different specialties, as in Poland the main role in vaccines administration attributes to family medicine doctors, paediatricians and internal medicine specialists. Considering the results obtained so far, we are planning to increase the number of physicians to be interviewed.